# Risk factors of HIV infection among female entertainment workers in Cambodia: Findings of a national survey

Sovannary Tuot[1], Alvin Kuo Jing Teo[2], Pheak Chhoun[1], Phalkun Mun[3], Kiesha Prem[2,4], Siyan Yi[1,2,5]*

1 KHANA Center for Population Health Research, Phnom Penh, Cambodia, 2 Saw Swee Hock School of Public Health, National University of Singapore and National University Health System, Singapore, Singapore, 3 National Center for HIV/AIDS, Dermatology and STD, Phnom Penh, Cambodia, 4 Department of Infectious Disease Epidemiology, Faculty of Epidemiology and Population Health, London School of Hygiene & Tropical Medicine, London, United Kingdom, 5 Center for Global Health Research, Touro University California, Vallejo, California, United States of America

* ephsyi@nus.edu.sg

## Abstract

### Background

Cambodia has been well recognized for its success in the fight against the HIV epidemic. However, challenges remain in eliminating HIV infections in key populations, including women working in entertainment establishments, such as massage parlors, karaoke bars, or beer gardens. This study explored the prevalence of HIV and identified factors associated with HIV infection among female entertainment workers (FEWs) in Cambodia.

### Methods

This national biological and behavioral survey was conducted in 2016 in Phnom Penh and 17 provinces. We used a two-stage cluster sampling method to recruit FEWs for HIV testing performed on-site and face-to-face interviews using a structured questionnaire. We investigated factors associated with HIV infection using multiple logistic regression.

### Results

This study included 3149 FEWs with a mean age of 26.2 years (SD 5.7). The adjusted prevalence of HIV was 3.2% (95% CI 1.76–5.75). In the multiple logistic regression model, the odds of HIV infection were significantly higher among FEWs in the age group of 31 to 35 (AOR 2.72, 95% CI 1.36–8.25) and 36 or older (AOR 3.62, 95% CI 1.89–10.55); FEWs who were not married but living with a sexual partner (AOR 3.00, 95% CI 1.16–7.79); FEWs who had at least ten years of formal education (AOR 0.32, 95% CI 0.17–0.83); FEWs who reported having abnormal vaginal discharge (AOR 3.51, 95% CI 1.12–9.01), genital ulcers or sores (AOR 2.06, 95% CI 1.09–3.17), and genital warts (AOR 2.89, 95% CI 1.44–6.33) in the past three months; and FEWs who reported using illicit drugs (AOR 3.28, 95% CI 1.20–4.27) than their respective reference group. The odds of HIV infection were significantly

**Data Availability Statement:** All relevant data are within the paper and its Supporting Information files.

**Funding:** This study was funded by the Global Fund to Fight AIDS, Tuberculosis, and Malaria through the National Center for HIV/AIDS, Dermatology, and STD. The funding was part of the GF-New Funding Model grant (GF-NFM 2015-2017). KHANA Center for Population Health Research provided technical support to the study design, data collection, data analyses, and report writing. The funders had no role in study design, data collection and analysis, decision to publish, or preparation of the manuscript. No authors received a salary from the funder.

**Competing interests:** The authors have declared that no competing interests exist.

lower among FEWs working in karaoke bars (AOR 0.26, 95% CI 0.14–0.50) and beer gardens (AOR 0.17, 95% CI 0.09–0.54) than among freelance FEWs.

## Conclusions

The prevalence of HIV among FEWs in Cambodia remains much higher than that in the general population. These findings indicate that differentiated strategies to address HIV and other sexually transmitted infections should be geared towards FEWs working as freelancers or in veiled entertainment venues such as massage parlors and freelance sex workers. Prevention efforts among venue-based FEWs should be sustained.

## Introduction

The national response to the human immunodeficiency virus (HIV) in Cambodia has been described as tri-phasic [1]. In the first phase (1991–2000), Cambodia faced one of the fastest-growing HIV epidemics in Asia; and in 1998, the HIV prevalence peaked at 1.7% among the general adult population aged 15 to 49 [2, 3]. The second phase (2001 to 2010) was characterized by combined efforts related to treatment, decentralization of services, and enhanced focus on key populations [4]. As a result of these strategies, the epidemic was significantly controlled. In recognition of the national success in controlling the HIV epidemic, Cambodia was feted with the United Nations' Millennium Development Goal award in 2010 [5, 6]. The current third phase (2011–2020) aims to achieve three-zero targets: (1) zero new HIV infection, (2) zero AIDS-related death, and (3) zero AIDS-related discrimination in communities [4]. Initial evidence suggests that the HIV epidemic is on a downward trajectory, and the HIV prevalence in the general adult population aged 15 to 49 declined to 0.6% in 2016 [7]. Due to the expanded access to antiretroviral therapy (ART), annual AIDS-related deaths have been reduced by two-thirds over the past ten years [1, 8].

Despite these tremendous successes, Cambodia has faced significant challenges in controlling the HIV epidemic in key populations, including female entertainment workers (FEWs). In Cambodia, FEWs refer to women working in entertainment establishments such as karaoke bars, massage parlors, or beer gardens. FEWs working in these establishments may or may not be involved in transactional sex. The FEW population also includes freelance sex workers who solicit clients in public places, such as streets or parks, or on calls. In 2008, the Cambodian government enacted a new Law to Suppress Human Trafficking and Sexual Exploitation [9], which has made health interventions in sites used for transactional sex more difficult. Due to the closure of many brothels, sex work has moved to entertainment-based venues or other informal and hidden settings like streets and parks. The distinction between direct and indirect sex work has become obscured, and an increase in indirect transactional relationships, such as sweethearts, have been documented [10]. In the Cambodian context, "sweethearts" refer to romantic non-commercial sexual relationships. For FEWs, a sweetheart is typically a boyfriend or regular client in the form of indirect transactional sex through living support, dinner dates, gifts, or shopping trips. Such relationships usually involve a lack of condom use as displays of trust and intimacy [11].

Among FEWs, the HIV prevalence is exceptionally high in sub-groups such as brothel-based (17.4%) and street-based (37.3%) sex workers than among women working in other entertainment establishments (9.8%) [7]. Couture et al. conducted a prospective study of young FEWs in the capital city of Phnom Penh and found that the HIV incidence among the

FEWs was 3.6 per 100 person-years (95% confidence interval [CI], 1.2%–11.1%) [12]. The identified risk factors of the HIV infection in the study included being freelance sex workers and younger age of first sex (≤15 years). Our recent study found that almost half of the FEW participants had not tested for HIV in the past six months preceding the survey [11]. Sexually transmitted infections (STIs), substance use, mental health problems, and gender-based violence were also commonly reported among FEWs in Cambodia [12–20], while the rates of consistent condom use remain consistently low [11, 21]. These conditions and behaviors have been reported as risk factors of HIV infection and HIV risk behaviors among this key population [12–17].

Collectively, these empirical data suggest that FEWs in Cambodia remain at disproportionally high risk of HIV infection. Due to the significant changes in the sex and entertainment industry landscapes, the information on the burden of HIV infection and its related risk factors needs to be closely followed up. The strategic information with up-to-date data on the HIV epidemic and programmatic response is crucial to Cambodia, particularly in the wake of the shrinking international HIV funding. The reduced funding is due to the country's status change from a low-income to a lower-middle-income country in 2016 [22]. More strategic decisions need to be made to adapt to the limited resources to achieve the ambitious global targets of ending the HIV epidemic by 2030. Therefore, we conducted this study to explore the HIV prevalence and identify risk factors associated with HIV infection among FEWs in Cambodia.

## Materials and methods

### Study sites and participants

This National Integrated Biological and Behavioral Survey among FEWs (FEW-IBBS) was conducted in 2016 in Phnom Penh and 17 other provinces purposively selected out of the 25 provinces in Cambodia. The FEW population size in the selected provinces represented more than 90% of the total FEWs in Cambodia in 2015 [7, 23]. The survey included women who were (1) working in the entertainment establishments or as freelance sex workers; (2) aged at least 18 years; (3) sexually active, defined as having had vaginal or anal sexual intercourse with at least one man in the past 12 months; (4) able to communicate in Khmer; (5) able and willing to provide written informed consent; and (6) willing to be physically present at the study site for an interview and HIV testing.

### Sample size calculation

We used the Open Epi calculator version 3.02 [24] to calculate the sample size required to estimate the HIV prevalence. In 2016, the estimated total number of FEWs in Cambodia was 40215, working in 2571 entertainment establishments, including 896 freelance FEWs [3, 7]. Using the HIV prevalence of 3.1% in the most recent national survey [23], a design effect of 1.5, 10% refusal rate, and a 95% confidence level, the minimum sample size required for the survey was 3100 FEWs.

### Sampling procedures

Venue-based FEWs were sampled using a multi-stage cluster sampling design. First, we listed all the entertainment establishments clusters (karaoke bars, massage parlors, beer gardens, and beer companies) in the selected areas and the estimated number of FEWs in each cluster. Next, we assigned a number to each cluster. We used a random number calculator to randomly select a cluster from the list. Subsequently, we invited all the FEWs in the selected clusters to

partake in the study. We repeated the random cluster selection until the target sample size was attained.

We used a time-location sampling method to recruit freelance FEWs. We established a sampling frame comprised of all known hotspots where freelance FEWs were known to assemble using information obtained from non-governmental organizations (NGOs) working with this population in the different localities. Before data collection, the study team worked closely with the respective NGOs to conduct a feasibility assessment and updated the sampling frame accordingly. We randomly selected the locations, date, and time (in four-hour time periods) for participant recruitment. We anticipated that the number of hotspots in each province would be small. Therefore, we included all hotspots and invited all FEWs found at the hotspots to participate in the study.

## Data collection training

In this national survey, we involved representatives of key stakeholders, including FEW communities, in every stage of the survey, from study design to questionnaire development and validation, data collection, and finding dissemination. Pre-data-collection workshops and training were conducted with data collection teams and representatives of HIV key stakeholders, NGOs working with FEWs, and FEW communities. Data collection teams were composed of team leaders from the Surveillance Unit of the National Center for HIV/AIDS, Dermatology and STD, staff members of the respective provincial health department, and representatives of NGOs, community outreach workers, and FEW communities. The training covered: (1) sampling methods, including mapping and eligibility criteria, (2) informed consent procedures, (3) participants' privacy and confidentiality, (4) study protocol and interview techniques, (5) blood specimen collection, and (6) record-keeping and completion of the survey forms. We also pretested the questionnaire during the training.

## Questionnaire development

We developed a structured questionnaire for face-to-face interviews, which took approximately 30 minutes to complete. Standardized and validated tools were adapted from previous studies among HIV key populations in Cambodia [18–20, 25, 26]. The structured questionnaire was initially developed in English and then translated into Khmer, Cambodia's national language. It was then back-translated into English by another translator to ensure that the original items' "content and spirit" were maintained. A consultative meeting was held with representatives of key stakeholders working on HIV and harm reduction programs, including NGOs and FEW communities, to review the study protocol and tools. A pilot study was conducted with 20 FEWs in Phnom Penh, later excluded from the main study.

## Variables and measurements

Socio-demographic characteristics included age (continuous), urbanicity of study sites, marital status, years of formal education attained (continuous), and type of entertainment venue for which the woman was working. We also collected information on living duration in the current city (continuous) and working duration in the current entertainment venue (continuous).

For HIV risks, we collected information regarding the participant's HIV status and sexual behaviors with commercial partners, defined as having sexual intercourse in exchange for money or gifts, and non-commercial partners. These included the number of sexual partners (continuous), condom use frequency (always, frequently, sometimes, never), STI diagnosis, and experiences of STI symptoms (yes or no) in the past three months. We also collected information on illicit drug use (heroin, marijuana, amphetamine-type stimulants, or other types of

drugs) and alcoholic drinks at work (yes or no) in the past three months. For participants who reported living with HIV, we collected self-reported HIV care and treatment information, including ART.

All participants received HIV testing regardless of whether they already knew their HIV-status. A blood sample was obtained from each participant by a trained laboratory technician through finger-prick and tested for HIV antibodies using the HIV-1/2 Determine™ test, following the national protocol [27]. We confirmed the test results by the HIV 1/2 STAT-PAK™ test on-site. If a specimen was reactive by the HIV-1/2 Determine™ test but non-reactive by the HIV 1/2 STAT-PAK™ test, the participant was recommended to go for a confirmatory test at an ART clinic of their choice with support from a community outreach worker. HIV test result was provided to the participants verbally after the interview, together with post-test counseling. All participants attended a pre- and post-test counseling session provided by trained counselors from HIV confidential counseling and testing centers located in the study sites. Those tested positive for HIV were referred to an ART clinic of their choice by a local NGO working in the respective area for care and treatment services according to the national guidelines. All participants received a token of appreciation valued at approximately US$4 for their time and effort.

### Statistical analyses

Sampling weights that corrected for non-response and sample design were applied. Standard errors were adjusted for clustering at the venue level [28]. We calculated the HIV prevalence by dividing the number of HIV-positive participants by the total number of participants tested. In bivariate analyses, we used the Chi-squared test (or Fisher's exact test when a cell count was smaller than five) for categorical variables and Student's $t$-test for continuous variables. A multiple logistic regression model was built to identify risk factors associated with HIV infection. We transformed the age and education level into categorical variables. We included age, education level, marital status, types of entertainment establishments, and other variables significantly associated with HIV infection in bivariate analyses ($p<0.05$) simultaneously in the model. We then removed variables not statistically significant from the model using a backward stepwise selection method. We reported adjusted odds ratios (AOR) with its associated 95% CI and $p$-values. Statistical analyses were conducted using STATA version 12.0 (Stata Corp, Texas, United States).

### Ethical considerations

The National Ethics Committee for Health Research, Ministry of Health, Cambodia approved the study (Ref no: 297NECHR). The data collection team explained to the participants about the study and obtained their written consent. Participation in the study was voluntary, and participants could refuse to respond to any questions or discontinue their participation at any time. We also extended free HIV testing to eligible individuals who refused to participate in this study if they wanted. We maintained the confidentiality of the participants by using unique codes, and no personal identifiers were recorded.

## Results

### HIV prevalence and socio-demographic characteristics

Of 3353 FEWs invited, 148 (4.4%) refused to participate in the study, primarily due to their time constraints. Fifty-six participants (1.8%) with missing data of main variables or HIV testing results were further excluded from the analyses. In total, we included 3149 FEWs in the

**Table 1. Socio-demographic characteristics of HIV-positive and HIV-negative FEWs (*n* = 3149).**

| Variables | | Total | HIV-negative | HIV-positive | P-value* |
|---|---|---|---|---|---|
| Age groups | | | | | <0.001 |
| | < 21 | 584 (18.5) | 581 (18.9) | 3 (4.2) | |
| | 21–25 | 1004 (31.9) | 994 (32.3) | 10 (13.9) | |
| | 26–30 | 905 (28.7) | 894 (29.1) | 11 (15.3) | |
| | 31–35 | 455 (14.4) | 423 (13.7) | 32 (44.4) | |
| | ≥ 36 | 201 (6.4) | 185 (6.0) | 16 (22.2) | |
| Current marital status | | | | | <0.001 |
| | Never married | 1072 (34.0) | 1061 (34.5) | 11 (15.3) | |
| | Married, living together | 548 (17.4) | 540 (17.5) | 8 (11.1) | |
| | Married, not living together | 113 (3.6) | 109 (3.5) | 4 (5.6) | |
| | Divorced/widowed | 1215 (38.6) | 1180 (38.3) | 35 (48.6) | |
| | Not married, living with a partner | 201 (6.4) | 187 (6.1) | 14 (19.4) | |
| Level of formal education attained | | | | | <0.001 |
| | ≤ 6 | 1894 (60.1) | 1834 (59.6) | 60 (83.3) | |
| | 7–9 | 958 (30.4) | 948 (30.8) | 10 (13.9) | |
| | ≥ 10 years | 297 (9.4) | 295 (9.6) | 2 (2.8) | |
| Type of entertainment venues | | | | | <0.001 |
| | Massage parlors | 93 (3.0) | 89 (2.9) | 4 (5.6) | |
| | Freelance sex workers | 351 (11.1) | 312 (10.1) | 39 (54.2) | |
| | Beer companies | 101 (3.2) | 99 (3.2) | 2 (2.8) | |
| | Karaoke bars | 2223 (70.6) | 2197 (71.4) | 26 (36.1) | |
| | Beer gardens | 381 (12.1) | 380 (12.3) | 1 (1.4) | |
| Duration of working for the current entertainment venue ≥ 12 months | | 1380 (43.9) | 1336 (43.4) | 44 (61.1) | 0.003 |

Abbreviations: FEW, female entertainment workers; HIV, human immunodeficiency virus.

*Chi-square test (or Fisher's exact test when a cell count was smaller than 5) was used.

analyses. The mean age of the participants was 26.2 years (SD 5.7). The adjusted HIV prevalence among FEWs in this study was 3.2%. The prevalence was 11.1% among freelance FEWs (11.1%) and 4.3% among FEWs working in massage parlors. Of those who knew their HIV-positive status (*n* = 29), 86.2% were on ART. As shown in Table 1, most of the participants were aged 30 or younger, and 38.6% were divorced or widowed. The participants' education level was generally low, with 60.1% having attained six years or less of formal education. More than two-thirds (70.6%) worked in karaoke bars, and 43.9% reported having worked in the current venue for more than one year. HIV infection was significantly associated with age groups, marital status, formal education level, entertainment venue type, and working duration in the current entertainment venue.

## HIV risk behaviors

Table 2 shows that 24.9% of women reported always using condoms with non-commercial partners in the past three months. Of the 1396 (53.0%) women who reported having sexual intercourse in exchange for money or gifts in the past 12 months, 19.5% reported having two or more commercial partners on the last working day, and 80.5% reported always using condoms with this type of partners in the past three months. Regarding substance use, 7.7% reported using illicit drugs in the past three months, with amphetamine-type stimulants being the most commonly used drugs. About two-thirds (62.6%) reported drinking alcohol at work

**Table 2. HIV risks among HIV-positive and HIV-negative FEWs in the study (*n* = 3149).**

| Sexual behaviors and STI symptoms | | Total | HIV-negative | HIV-positive | P-value* |
|---|---|---|---|---|---|
| Condom use with non-commercial partners in the past 3 months (*n* = 1473) | | | | | <0.001 |
| | Always | 367 (24.9) | 349 (24.2) | 18 (56.3) | |
| | Frequently | 67 (4.5) | 66 (4.6) | 1 (3.1) | |
| | Sometimes | 245 (16.7) | 241 (16.7) | 4 (12.5) | |
| | Never | 794 (53.9) | 785 (54.5) | 9 (28.1) | |
| Had sex in exchange for money or gifts in the past 12 months | | 1396 (53.0) | 1343 (52.3) | 53 (77.9) | <0.001 |
| Had ≥2 commercial sex partners on last working day (*n* = 1396) | | 272 (19.5) | 254 (18.9) | 18 (34.0) | 0.007 |
| Condom use with commercial sexual partners in the last 3 month (*n* = 1292) | | | | | 0.12 |
| | Always | 1040 (80.5) | 996 (80.3) | 44 (86.3) | |
| | Frequently | 95 (7.4) | 93 (7.5) | 2 (3.9) | |
| | Sometimes | 113 (8.7) | 109 (8.8) | 4 (7.8) | |
| | Never | 44 (3.4) | 43 (3.5) | 1 (2.0) | |
| Forced sex in the past 3 months | | 42 (1.4) | 36 (1.2) | 6 (8.3) | <0.001 |
| Gang rape in the past 3 months | | 40 (1.3) | 35 (1.1) | 5 (6.9) | <0.001 |
| Illicit drug use in the past 3 months | | 242 (7.7) | 216 (7.0) | 26 (36.1) | <0.001 |
| | ATS use | 205 (86.1) | 181 (85.4) | 24 (92.3) | 0.55 |
| | Injecting drug use | 19 (8.1) | 16 (7.7) | 3 (11.5) | 0.45 |
| Drinking alcohol at work every day | | 1970 (62.6) | 1941 (63.1) | 29 (40.3) | <0.001 |
| Having ≥5 alcoholic drinks on the last working day | | 2545 (80.8) | 2500 (81.2) | 45 (62.5) | <0.001 |
| Having been diagnosed with an STI in the past 3 months | | 635 (20.2) | 613 (19.9) | 22 (30.6) | 0.03 |
| STI symptoms in the past 3 months | | | | | |
| | Abnormal vaginal discharge | 1066 (33.9) | 1033 (33.6) | 33 (45.8) | 0.03 |
| | Lower abdominal pain | 814 (25.8) | 785 (25.5) | 29 (40.3) | 0.005 |
| | Genital ulcers or sores | 78 (2.5) | 72 (2.3) | 6 (8.3) | 0.001 |
| | Genital warts | 42 (1.3) | 38 (1.2) | 4 (5.6) | 0.02 |

Abbreviations: ATS, amphetamine-type stimulants; FEW, female entertainment workers; HIV, human immunodeficiency virus; IBBS, integrated biological and behavioral survey.

*Chi-square test (or Fisher's exact test when a cell count was smaller than 5) was used.

every day, and 80.8% reported having five or more alcoholic drinks on their last working day. About one in five (20.2%) reported having been diagnosed with an STI in the past three months. The most common facilities where the participants received the STI diagnosis were public health facilities (55.4%) and NGOs' clinics (34.7%). The participants also reported different forms of STI symptoms experienced in the past three months, including abnormal vaginal discharge (33.9%), lower abdominal pain (25.8%), ulcers or sores in genital areas (2.5%), and genital warts (1.3%). HIV infection was significantly associated with consistent condom use levels with non-commercial partners, involvement in transactional sex, the number of commercial sex partners, and forced sex and gang-rape experience. HIV infection was also significantly associated with STI diagnosis and symptoms and substance use, including illicit drug use, alcohol use, and binge drinking.

## Risk factors of HIV infection

Table 3 shows the results of the multiple logistic regression analysis. After controlling for other covariates, the odds of HIV infection were significantly higher among FEWs in the age group

**Table 3. Factors associated with HIV infection among female entertainment workers (*n* = 3149).**

| Variables in the final model | | AOR (95% CI) | P-value* |
|---|---|---|---|
| Age group | | | |
| | < 21 | Reference | |
| | 21–25 | 1.62 (0.43–5.20) | 0.48 |
| | 26–30 | 1.81 (0.47–4.05) | 0.39 |
| | 31–35 | 2.72 (1.36–8.25) | 0.001 |
| | ≥ 36 | 3.62 (1.89–10.55) | 0.004 |
| Current marital status | | | |
| | Never married, not living with a partner | Reference | |
| | Not married, living with a partner | 3.00 (1.16–7.79) | 0.02 |
| | Married, living together | 0.56 (0.20–1.53) | 0.25 |
| | Married, not living together | 2.17 (0.62–7.63) | 0.23 |
| | Divorced/widowed | 1.24 (0.56–2.74) | 0.60 |
| Level of formal education attained | | | |
| | ≤ 6 | Reference | |
| | 7–9 | 0.74 (0.36–1.52) | 0.41 |
| | ≥10 years | 0.32 (0.17–0.83) | 0.02 |
| Type of entertainment venues | | | |
| | Freelance sex workers | Reference | |
| | Massage parlors | 0.91 (0.28–2.92) | 0.87 |
| | Beer companies | 0.34 (0.07–1.60) | 0.17 |
| | Karaoke bars | 0.26 (0.14–0.50) | <0.001 |
| | Beer gardens | 0.17 (0.09–0.54) | 0.01 |
| Illicit drug use in the past 3 months | | | |
| | No | Reference | |
| | Yes | 3.28 (1.20–4.27) | 0.002 |
| Had abnormal vaginal discharge in the past 3 months | | | |
| | No | Reference | |
| | Yes | 3.51 (1.12–9.01) | 0.01 |
| Had genital ulcer or sores in the past 3 months | | | |
| | No | Reference | |
| | Yes | 2.06 (1.09–3.17) | 0.02 |
| Had genital warts in the past 3 months | | | |
| | No | Reference | |
| | Yes | 2.89 (1.44–6.33) | 0.006 |

Abbreviations: AOR, adjusted odds ratio; CI, confidence interval; FEW, female entertainment workers; HIV, human immunodeficiency virus.

* Age, marital status, education level, entertainment venue, and variables associated with HIV infection in the bivariate analyses at a level of p<0.05 were simultaneously included in the model.

of 31 to 35 (AOR 2.72, 95% CI 1.36–8.25) and 36 or older (AOR 3.62, 95% CI 1.89–10.55) than those in the age group of <21 years. The odds of HIV infection were significantly higher among FEWs who were not married but living with a partner than those who were never married and not living with a partner (AOR 3.00, 95% CI 1.16–7.79). The odds of HIV infection were significantly lower among FEWs who had attained at least ten years of formal education than those attaining ≤six years of formal education (AOR 0.32, 95% CI 0.17–0.83). Compared to freelance FEWs, the odds of HIV infection were significantly lower among FEWs working

at karaoke bars (AOR 0.26, 95% CI 0.14–0.50) and beer gardens (AOR 0.17, 95% CI 0.09–0.54). The odds of HIV infection were significantly higher among FEWs who reported having abnormal vaginal discharge (AOR 3.51, 95% CI 1.12–9.01), genital ulcers or sores (AOR 2.06, 95% CI 1.09–3.17), and genital warts (AOR 2.89, 95% CI 1.44–6.33) in the past three months than those who did not. The odds of HIV infection were also significantly higher among FEWs who reported using illicit drugs (AOR 3.28, 95% CI 1.20–4.27) than those who did not.

## Discussion

We found that the HIV prevalence among FEWs in this national survey was 3.2%, about six times higher than the estimated 0.6% prevalence among the Cambodian general population [4] and 0.3% among Cambodian pregnant women attending antenatal care in the same year [7]. This study also identified several risk factors associated with HIV infection among FEWs in this national survey. Our results showed that the risk of HIV infection increased significantly with age and was associated with the nature of entertainment venues where FEWs were working. The prevalence was also significantly higher among FEWs with histories of illicit drug use and STI diagnosis and symptoms such as genital warts, genital ulcers or sores, and abnormal vaginal discharge. The risk of HIV infection was inversely associated with years of formal education attained.

We found that the HIV prevalence was exceptionally high among specific subgroups of FEWs, such as freelance FEWs (11.1%) and FEWs working in massage parlors (4.3%). Past surveys have also indicated a high HIV prevalence among various groups of FEWs, such as brothel-based (17.4%) and street-based (37.3%) sex workers, as compared to the prevalence among women working in other entertainment establishments (9.8%) [12]. In the absence of a safer working environment such as brothels [12, 17, 29], our findings suggest that differentiated programming is needed to ensure that HIV prevention efforts are intensified, particularly for street-based and freelance FEWs.

The higher prevalence of HIV among FEWs in the older age groups found in this study is in line with findings from another study that reported a similar relationship between HIV infection and older FEWs in Cambodia [30]. The cumulative exposure to HIV risks could explain this association. Previous studies in Cambodia also suggested that street-based FEWs tend to be older than women working in brothels and entertainment venues [12, 14, 17]. In this study, most HIV positive cases detected in the older age groups (>30 years) were aware of their HIV status before the survey.

This study found that FEWs who were not married but living with their partners were at a higher risk of HIV. We also observed a similar trend among FEWs who were married but not living with their partners. This observation is consistent with findings from previous studies in Cambodia, which showed low rates of consistent condom use in non-commercial relationships among FEWs [11, 19, 20]. In our recent study, unprotected sex was reported by FEWs to be a way to express trust and faithfulness to their regular non-commercial partners (sweethearts) [11, 19]. Together, these findings may explain the higher risk of HIV among non-married FEWs living with partners. Given the consistency of this observation, new strategies to increase consistent condom use among FEWs are warranted.

Besides, our results indicated that the risk of HIV was higher among FEWs who reported illicit drug use. This finding could be attributed to the negative impact of illicit drug use on consistent condom use, numbers of sex partners, or unsafe injection practices [12, 31–33]. Furthermore, our study found that the presence of STI symptoms was associated with HIV infection, and it is consistent with other evidence regarding the relationship between STIs and HIV infection [12, 16]. The high prevalence of STI symptoms observed in our study is congruous

with findings from other studies conducted among FEWs in Cambodia [11, 12, 14, 19]. These results suggested that tailored education efforts should be implemented, emphasizing regular condom use, condom negotiation skills, and the detrimental effects of alcohol and other substance use on HIV exposure. Risk-reduction education is crucial, given that the prevalence of HIV was higher among less educated FEWs.

## Strengths and limitations

The strengths of this study include the implementation of multi-stage sampling procedures to engage a large sample of FEWs across sites with a high burden of HIV in Cambodia. We used standardized data collection procedures and validated tools to collect biological samples and survey data across all study sites. Furthermore, this survey involved pertinent stakeholders at different national health system levels, NGOs, and community members in developing the study protocol, tools, and strategies for disseminating study findings.

Despite these strengths, this study also has several limitations. First, this study did not include seven provinces with a lower burden of HIV and a smaller FEW population. Therefore, the study findings may not be generalized to a national level, although the data were appropriately weighted in the analyses. Second, we collected data on sensitive issues such as sexual behaviors using self-reported measures through face-to-face interviews that may result in potential social desirability bias. The risks are likely to be underestimated, given the cultural norms governing sexual behaviors and substance use in Cambodia. Third, albeit minimal, the monetary incentive given to the participants and their connection to the ongoing community-based HIV programs may have affected their genuine motivation to partake in the study and potentially influenced their response to the survey. Nevertheless, we believe that we took sufficient measures to minimize these potential effects throughout the data collection. Finally, we could not meet the sample size requirement for freelance FEWs due to difficulty reaching this population. Therefore, generalizability to the entire population of freelance FEWs could be limited.

## Conclusions

This study documents the prevalence and risk factors of HIV infection among FEWs in Cambodia. The results showed that the HIV infection risk was associated with older age, the nature of the entertainment venues where FEWs were working, and history of illicit drug use and STI symptoms in the past three months. The risk of HIV infection was inversely associated with years of formal education attained. These findings indicate that differentiated strategies for HIV prevention among FEWs should be geared towards the FEWs working as freelancers or in veiled entertainment venues, such as massage parlors, and freelance FEWs. However, prevention efforts for venue-based FEWs should be maintained. Innovative interventions such as online services and mobile health technologies using text and voice messaging may be more useful than the current reliance on physical outreach activities to effectively reach freelance and other high-risk FEWs for early HIV detection and linking them to treatment and care services.

## Supporting information

**S1 Questionnaire. (English).**
(PDF)

**S2 Questionnaire. (Khmer).**
(PDF)

## Acknowledgments

This study was conducted in a collaboration between the National Center for HIV/AIDS, Dermatology and STD and the consortium partners of the HIV/AIDS Flagship Project, including KHANA, FHI360, PSI/PSK. We thank all implementing partners and participants in the study who fully supported the study. Special thanks go to Janika Sullivan, a master's student from the Public Health Program, Touro University California, USA and an international intern at KHANA Center for Population Health Research for her excellent inputs in the manuscript development.

## Author Contributions

**Conceptualization:** Sovannary Tuot, Pheak Chhoun, Phalkun Mun, Kiesha Prem, Siyan Yi.

**Data curation:** Sovannary Tuot, Alvin Kuo Jing Teo, Phalkun Mun, Kiesha Prem, Siyan Yi.

**Formal analysis:** Sovannary Tuot, Alvin Kuo Jing Teo, Kiesha Prem, Siyan Yi.

**Funding acquisition:** Sovannary Tuot, Phalkun Mun, Siyan Yi.

**Investigation:** Sovannary Tuot, Alvin Kuo Jing Teo, Pheak Chhoun, Phalkun Mun, Kiesha Prem, Siyan Yi.

**Methodology:** Sovannary Tuot, Alvin Kuo Jing Teo, Phalkun Mun, Kiesha Prem, Siyan Yi.

**Project administration:** Sovannary Tuot, Pheak Chhoun, Phalkun Mun, Siyan Yi.

**Resources:** Sovannary Tuot, Pheak Chhoun, Phalkun Mun, Siyan Yi.

**Software:** Sovannary Tuot, Alvin Kuo Jing Teo, Kiesha Prem, Siyan Yi.

**Supervision:** Sovannary Tuot, Pheak Chhoun, Phalkun Mun, Siyan Yi.

**Validation:** Sovannary Tuot, Alvin Kuo Jing Teo, Pheak Chhoun, Phalkun Mun, Kiesha Prem, Siyan Yi.

**Visualization:** Sovannary Tuot, Alvin Kuo Jing Teo, Kiesha Prem, Siyan Yi.

**Writing – original draft:** Sovannary Tuot, Alvin Kuo Jing Teo, Kiesha Prem, Siyan Yi.

**Writing – review & editing:** Sovannary Tuot, Alvin Kuo Jing Teo, Pheak Chhoun, Phalkun Mun, Kiesha Prem, Siyan Yi.

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
