## [Decision Letter · Decision Letter 0]

23 Oct 2020

PONE-D-20-22691

Risk factors of HIV infection among female entertainment workers in Cambodia

PLOS ONE

Dear Dr. Yi,

Thank you for submitting your manuscript to PLOS ONE. After careful consideration, we feel that it has merit but does not fully meet PLOS ONE’s publication criteria as it currently stands. Therefore, we invite you to submit a revised version of the manuscript that addresses the points raised during the review process.

We look forward to receiving your revised manuscript.

Kind regards,

Petros Isaakidis

Academic Editor

PLOS ONE

Journal Requirements:

2) Please include a copy of the interview guide used in the study, in both the original language and English, as Supporting Information, or include a citation if it has been published previously."

3) Please state whether you validated the questionnaire prior to testing on study participants. Please provide details regarding the validation group within the methods section.

4) We note that you have indicated that data from this study are available upon request. PLOS only allows data to be available upon request if there are legal or ethical restrictions on sharing data publicly. For information on unacceptable data access restrictions, please see http://journals.plos.org/plosone/s/data-availability#loc-unacceptable-data-access-restrictions.

5) Thank you for stating the following financial disclosure:

 [The funders had no role in study design, data collection and analysis, decision to

publish, or preparation of the manuscript].

Reviewers' comments:

Reviewer's Responses to Questions

**Comments to the Author**

1. Is the manuscript technically sound, and do the data support the conclusions?

Reviewer #1: Partly

Reviewer #2: Partly

2. Has the statistical analysis been performed appropriately and rigorously? 

Reviewer #1: I Don't Know

Reviewer #2: Yes

3. Have the authors made all data underlying the findings in their manuscript fully available?

Reviewer #1: Yes

Reviewer #2: Yes

4. Is the manuscript presented in an intelligible fashion and written in standard English?

Reviewer #1: Yes

Reviewer #2: Yes

5. Review Comments to the Author

Reviewer #1: Very clear study findings yet quite simple and predictable results for those who work with this topic and sector i.e street based SWs (or FEWs) are generally more vulnerable at risk to structural violence, which has direct link with higher HIV risks and prevalence. HIV prevalence is greater in the above 30 years of age etc

While the data supports conclusion, the low recruitment of freelance SWs studied would bring bias the results

Some bias may have also been introduced by the data collectors as they are not part of the Peer FEW community, and no mentioned of sensitization to this stigmatized population.

The acronym SW (sex worker) and using FEW (female entertainment worker) instead may confuse the identity of the studied population to the wider sector reader.

Would have expected to see violence (SGBV) as a risk factor for HIV in Sws , as may miss a significant opportunity to better guide interventions addressing the needs of FSWs, and in particular those whoa re 'freelance'

would have been good to understand where do most FSWs receive a 'diagnosis' of STIs? is this using SAM or lab tests at a health facility?

I would recommend to expand the referencing bibliography from similar contexts where SW occurs . Not sure if an article (No 26) on MSM is relevant

Abortion is mentioned as a contributing risk factor to HIV infection, not sure what can be infered form this, unless it is in discussion with inconsistent condom and contraceptive use . SRHR aspect could be further expanded.

Reviewer #2: The authors of the study report result of a HIV prevalence survey among female entertainment workers in Cambodia. The results show a higher prevalence of HIV-infection among this vulnerable population. The authors conclude by recommending that differentiated services should be available to these population groups based on their vulnerability assessment.

This study is very interesting, and I would like to congratulate the authors on providing evidence on the needs of this vulnerable population. However, there are aspects which need more clarity.

Major comments

1. In the study sites and participants section, the authors describe about the FEW-IBBS 2016 survey. While in the variables section, the authors mention that they developed a structured questionnaire. It is not clear from the methods section if the current findings come from the IBBS survey or from a separate survey. While most elements of the study design and questionnaire development are present throughout the manuscript, it will be easier for the readers if the authors can describe the design for the survey from which they are reporting their current findings in a dedicated study design section.

2. While the authors give a background on the study setting, especially about the work of FEWs, there are some aspects which are not clear, especially for readers who are unfamiliar with the Cambodian setting. From the definition of FEWs in the article, it seems that all of them are involved in transactional sex. However, in the results, the authors report that 53% had transactional sex in the past one year. This piece of information also goes against the inclusion criteria (line 116), which mentions that only those who had sex at least once in the past 12 months were included. If this definition included both transactional and non-transactional sex, then I am not clear how the risk due to the participants work as FEW is defined.

3. The authors use the term “sweetheart” in the manuscript. The way it is defined, I understand “sweetheart” as something akin to a “sugar daddy”. However, given the context of the study, it is not clear how “sweetheart” and FEWs are connected. This may be due to my inability to understand the terms being used in a Cambodian context. Hence, for clarity this needs more description. While “sweethearts” are mentioned in the introduction and discussion, they are not mentioned in the results. It is also not clear; how did the authors distinguish between “sweethearts” and FEWs engaged in commercial sex.

On a related note, in the discussion (lines 306-307), the authors mention, ‘unprotected sex was reported by the participants to be a way to express trust and faithfulness to their regular non-commercial partners (sweethearts)’. This finding is not found in the results section and appears for the first time in the discussion section. It is also not clear from the study methodology how did the authors arrive at this finding.

4. The authors mention that they conducted HIV testing for those who participated in the survey. However, it is not clear what happened with those who already knew their HIV status. Were they retested for this study? More information on this needs to reported in the methods and results section. The only clue regarding this comes from lines 301-303 in the discussion section, where the authors mention that most participants older than 30 years knew their HIV status. Information about ART status of those who knew their status will also be helpful.

5. The authors mention about desirability bias in the limitations section. They also mention that steps were taken to mitigate desirability bias. However, what steps were taken has not been mentioned. The only mention about this comes from line 340, which says “sufficient measure”. It is important to understand this, since, from the manuscript text, it seems that the surveyors, or at least the organisations that they worked for were known to the participants and the participants received some kind of services from them. This can affect participants’ response and hence it is important to explicitly mention what measures were taken to avoid this bias. It will also be helpful to mention who the surveyors were and how were they trained to avoid these biases. It would also be pertinent to mention how were the survey procedures were monitored.

6. The authors report a higher odds of HIV among those with STI. However, the model used does not contain condom use frequency. Could this result be due to the fact that those with STIs have infrequent condom use, which could also be the independent risk factor for HIV?

7. In sample size calculations, the authors seem to have calculated the overall sample size required for inclusion in the survey. However, in the limitations (line 342), the authors mention that they were unable to reach the sample size for freelance FEWs. Were different sample sizes for different subgroups calculated?

8. In the methods section (line 179), the authors mention that participants received a, ‘gift card not exceeding 4USD’. Did participants receive different amounts? If yes, what was the criteria for who received what amount?

9. In 348, the authors use the term “recent history of STI symptoms”. While the definition of ‘recent’ is not mentioned in the methods, from the results section we understand that it is three months. Please mention this definition, along with any other relevant operational definitions, in the methods section. Also, please rethink the use of the word ‘recent’ in the discussion.

10. In the conclusion, the authors recommend differentiated preventive services for FEWs in older age group since they had a higher odds of living with HIV than the lower age groups. I would request the authors to re-think about this. While it is true that the results show that FEWs had a higher proportion of those living with HIV. However, we do not know from the results how long ago did they test positive for HIV. In light of this, it is necessary that preventive measures should not be focussed only on the older age groups. It is necessary to ensure that preventive and testing services should be available from even the youngest age groups, since in this way, they might be able to effect behavioural change early, which might help prevent new HIV infections.

11. The authors mention that they found lower proportion of FEWs testing HIV-positive at the establishments. They also mention that, “street based FEWs tend to be older than those working in brothels and entertainment avenues”. Are these two facts related? Could it be that the entertainment avenues prefer to employ those who are younger? Or is it that if someone tests HIV-positive, then they might not be allowed to work at these entertainment avenues? Please revisit the related conclusion based on the answers to these questions.

12. The authors mention in the strengths that they had involved all stakeholders. It is very important during this kind of work to involve all the stakeholders from the beginning and I would like to congratulate the authors on this. It would be nice to understand a little more on what was done.

13. In line 91-93, the authors mention, ‘The identified risk factors of the HIV infection in the study included being freelance sex workers and younger age of first sex (≤15 years)’. However, this factor seems missing from this study. Was this variable collected and analysed?

Minor comments

1. Line 63 – The authors mention about a triphasic response. While, the second and third phases are clearly mentioned, the first phase is not clearly demarcated. It would be easier to read if the first phase is also clearly labelled so.

2. Line 113 – It would be good to also mention the total number of provinces for readers who are not aware about Cambodian setting.

3. Line 203 – It might be pertinent to report on how many participants refused to participate.

4. The tables as they stand are not very clear which makes them difficult to read. Please format them in a manner so that the subgroups and the totals are easily readable. Also, using row percent might make the tables easier to interpret, especially for tables 1 and 2.

6. PLOS authors have the option to publish the peer review history of their article (what does this mean?). If published, this will include your full peer review and any attached files.

Reviewer #1: **Yes: **Lucia O'Connell

Reviewer #2: No

---

## [Author Response · Author response to Decision Letter 0]

4 Dec 2020

Journal Requirements 

1. Please ensure that your manuscript meets PLOS ONE's style requirements, including those for file naming. The PLOS ONE style templates can be found at:

RESPONSE: We have formatted our manuscript and all supporting documents in accordance with PLOS ONE’s style requirements.

2. Please include a copy of the interview guide used in the study, in both the original language and English, as Supporting Information, or include a citation if it has been published previously." 

RESPONSE: We have included a copy of the questionnaire (in Khmer and English) as supplementary information.

3. Please state whether you validated the questionnaire prior to testing on study participants. Please provide details regarding the validation group within the methods section. 

RESPONSE: A structured questionnaire was developed based on similar previous national surveillance surveys among female entertainment workers and our recent studies. These studies are routinely conducted by the national HIV program to observe the trend of the HIV prevalence and risk behaviors among key populations in Cambodia using standardized methods and tools. We have added the following information in a new sub-section, ‘Questionnaire development,’ (Lines 165-174):

We developed a structured questionnaire for face-to-face interviews, which took approximately 30 minutes to complete. Standardized and validated tools were adapted from previous studies among HIV key populations in Cambodia [18-20,25,26]. The structured questionnaire was initially developed in English and then translated into Khmer, Cambodia’s national language. It was then back-translated into English by another translator to ensure that the original items’ “content and spirit” were maintained. A consultative meeting was held with representatives of key stakeholders working on HIV and harm reduction programs, including NGOs and FEW communities, to review the study protocol and tools. A pilot study was conducted with 20 FEWs in Phnom Penh, later excluded from the main study.

We note that you have indicated that data from this study are available upon request. PLOS only allows data to be available upon request if there are legal or ethical restrictions on sharing data publicly. For information on unacceptable data access restrictions, please see http://journals.plos.org/plosone/s/data-availability#loc-unacceptable-data-access-restrictions.

RESPONSE: We have received permission from the National Center for HIV/AIDS, Dermatology and STD and uploaded the anonymized data set as a Supplementary Material file.

[The funders had no role in study design, data collection and analysis, decision to

publish, or preparation of the manuscript].

a. Please clarify the sources of funding (financial or material support) for your study. List the grants or organizations that supported your study, including funding received from your institution.

d. If you did not receive any funding for this study, please state: “The authors received no specific funding for this work.”

RESPONSE: To address the queries, we have included the following amended statements in the cover letter:

a. This study was funded by the Global Fund to Fight AIDS, Tuberculosis, and Malaria through the National Center for HIV/AIDS, Dermatology, and STD. The funding was part of the GF-New Funding Model grant (GF-NFM 2015-2017). KHANA Center for Population Health Research provided technical support to the study design, data collection, data analyses, and report writing.

c. No authors received a salary from the funder. 

d. The authors received no specific funding for this work.

6. We noticed you have some minor occurrence of overlapping text with the following previous publication(s), which needs to be addressed:

https://bmcpublichealth.biomedcentral.com/track/pdf/10.1186/s12889-016-2814-6?site=bmcpublichealth.biomedcentral.com

In your revision ensure you cite all your sources (including your own works), and quote or rephrase any duplicated text outside the methods section. Further consideration is dependent on these concerns being addressed. 

RESPONSE: Thank you for the comment. We apologize for the oversight. We have rephrased the duplicated texts accordingly. 

Reviewer 1 

Very clear study findings yet quite simple and predictable results for those who work with this topic and sector i.e street based SWs (or FEWs) are generally more vulnerable at risk to structural violence, which has direct link with higher HIV risks and prevalence. HIV prevalence is greater in the above 30 years of age etc.

While the data supports conclusion, the low recruitment of freelance SWs studied would bring bias the results. 

RESPONSE: We thank the reviewer for your time reviewing our manuscript and providing constructive comments for improving it. We acknowledge the limitation due to the low recruitment rate of freelance sex workers. We have addressed all the reviewer’s comments and revised the manuscript accordingly. Please find our point-by-point response to your comments below.

Some bias may have also been introduced by the data collectors as they are not part of the Peer FEW community, and no mentioned of sensitization to this stigmatized population. 

RESPONSE: Thank you for raising these essential comments. In this national survey, we involved representatives of key stakeholders, including FEW communities, in every stage of the survey (study design, tool development and validation, data collection, findings dissemination). We have provided further information regarding the sensitization and preparation for the data collection in ‘Data collection training’ as follows (Lines 151-163):

In this national survey, we involved representatives of key stakeholders, including FEW communities, in every stage of the survey, from study design to questionnaire development and validation, data collection, and finding dissemination. Pre-data-collection workshops and training were conducted with data collection teams and representatives of HIV key stakeholders, NGOs working with FEWs, and FEW communities. Data collection teams were composed of team leaders from the Surveillance Unit of the National Center for HIV/AIDS, Dermatology and STD, staff members of the respective provincial health department, and representatives of NGOs, community outreach workers, and FEW communities. The training covered: (1) sampling methods, including mapping and eligibility criteria, (2) informed consent procedures, (3) participants’ privacy and confidentiality, (4) study protocol and interview techniques, (5) blood specimen collection, and (6) record-keeping and completion of the survey forms. We also pretested the questionnaire during the training.

The acronym SW (sex worker) and using FEW (female entertainment worker) instead may confuse the identity of the studied population to the wider sector reader. 

RESPONSE: We thank you for pointing out this issue. We understand that using the acronyms (SW and FEWs) may confuse the identity of the study population. Female entertainment workers are unique to HIV programs in Cambodia. They are women working in entertainment establishments, and many (but not all) are involved in sex work. Our previous studies found that approximately half of the female entertainment workers reported transactional sex in the past three months preceding the surveys. Therefore, it is fair to call them female entertainment workers, rather than sex workers. To minimize the confusion, we have defined female entertainment workers and revised the descriptions of female sex workers (from the literature) and female entertainment workers in this study where applicable. Please see Lines 78-81:

In Cambodia, FEWs refer to women working in entertainment establishments such as karaoke bars, massage parlors, or beer gardens. FEWs working in these establishments may or may not be involved in transactional sex. The FEW population also includes freelance sex workers who solicit clients in public places, such as streets or parks, or on calls.

Would have expected to see violence (SGBV) as a risk factor for HIV in Sws, as may miss a significant opportunity to better guide interventions addressing the needs of FSWs, and in particular those who are 'freelance' 

RESPONSE: We agreed that GBV is a critical issue and can be a risk factor for HIV infection among female sex and entertainment workers. In our recent studies, we have reported high exposure to GBV and its relationships with HIV risks, substance abuse, and mental health in this population. Unfortunately, we did not collect information on GBV in this national biological and behavioral survey to accommodate more competitive questions required for addressing the goal and objectives of the survey. 

Would have been good to understand where do most FSWs receive a 'diagnosis' of STIs? is this using SAM or lab tests at a health facility? 

RESPONSE: STI symptoms and diagnosis were self-reported. From our data, the most common facilities where the participants received STI diagnosis were public health facilities (55.4%) and NGOs’ clinics (34.7%). These facilities follow the national guidelines. We have included these figures in the Results section (Lines 255-257).

I would recommend to expand the referencing bibliography from similar contexts where SW occurs. Not sure if an article (No 26) on MSM is relevant 

RESPONSE: We thank the reviewer for this comment and agree that the referencing bibliography should be expanded within the contexts of female sex and entertainment workers. We have included relevant references where suitable. Reference No 26 (on MSM) was our previous national biological and behavioral survey among men who have sex with men, from which relevant questions on HIV risks (sexual behaviors, substance use, access to HIV services) were adapted. 

Abortion is mentioned as a contributing risk factor to HIV infection, not sure what can be infered form this, unless it is in discussion with inconsistent condom and contraceptive use . SRHR aspect could be further expanded. 

RESPONSE: This is an excellent comment. To avoid confusion, we have removed the mention of abortion as a contributing risk factor to HIV infection. We have also expanded the discussion on sexual and reproductive health and rights as suggested where relevant.

Reviewer 2 

General comments:

The authors of the study report result of a HIV prevalence survey among female entertainment workers in Cambodia. The results show a higher prevalence of HIV-infection among this vulnerable population. The authors conclude by recommending that differentiated services should be available to these population groups based on their vulnerability assessment.

This study is very interesting, and I would like to congratulate the authors on providing evidence on the needs of this vulnerable population. However, there are aspects which need more clarity. 

RESPONSE: We thank the reviewer for the encouraging comments about the contribution of this study to the body of knowledge on the needs of this vulnerable population. 

Major comments 

1. In the study sites and participants section, the authors describe about the FEW-IBBS 2016 survey. While in the variables section, the authors mention that they developed a structured questionnaire. It is not clear from the methods section if the current findings come from the IBBS survey or from a separate survey. While most elements of the study design and questionnaire development are present throughout the manuscript, it will be easier for the readers if the authors can describe the design for the survey from which they are reporting their current findings in a dedicated study design section. 

RESPONSE: We apologize for the confusion caused by our unclear description in the methods section. We also thank the reviewer for the opportunity to clarify. Data used for this study were collected in the national biological and behavioral survey among female entertainment workers (FEW-IBBS 2016). We developed a structured questionnaire for the FEW-IBBS 2016. The authors of this manuscript were the core team members of the survey. We have revised several parts in the Materials and Methods section and believe it is now clearer. 

2. While the authors give a background on the study setting, especially about the work of FEWs, there are some aspects which are not clear, especially for readers who are unfamiliar with the Cambodian setting. From the definition of FEWs in the article, it seems that all of them are involved in transactional sex. However, in the results, the authors report that 53% had transactional sex in the past one year. This piece of information also goes against the inclusion criteria (line 116), which mentions that only those who had sex at least once in the past 12 months were included. If this definition included both transactional and non-transactional sex, then I am not clear how the risk due to the participants work as FEW is defined. 

RESPONSE: We agree that the contexts of female entertainment workers can be confusing, especially for readers who are unfamiliar with HIV programs in the Cambodian setting. Differentiated from female sex workers, female entertainment workers are women working in different entertainment venues who may or may not sell sex. However, the umbrella population also covers female sex workers, mostly freelance or street-based sex workers, since brothel-based sex work has become illegal under a new law to suppress human trafficking and sexual exploitation. Our previous studies have consistently shown that approximately half of the female entertainment workers are involved in transactional sex. We have revised the definition of female entertainment workers as follows (Lines 78-81):

In Cambodia, FEWs refer to women working in entertainment establishments such as karaoke bars, massage parlors, or beer gardens. FEWs working in these establishments may or may not be involved in transactional sex. The FEW population also includes freelance sex workers who solicit clients in public places, such as streets or parks, or on calls.

We included the third inclusion criteria to exclude female entertainment workers who were not sexually active. It stated ‘sexually active, defined as having had vaginal or anal sexual intercourse with at least one man in the past 12 months,’ which referred to both transactional and non-transactional sex.

3. The authors use the term “sweetheart” in the manuscript. The way it is defined, I understand “sweetheart” as something akin to a “sugar daddy”. However, given the context of the study, it is not clear how “sweetheart” and FEWs are connected. This may be due to my inability to understand the terms being used in a Cambodian context. Hence, for clarity this needs more description. While “sweethearts” are mentioned in the introduction and discussion, they are not mentioned in the results. It is also not clear; how did the authors distinguish between “sweethearts” and FEWs engaged in commercial sex.

On a related note, in the discussion (lines 306-307), the authors mention, ‘unprotected sex was reported by the participants to be a way to express trust and faithfulness to their regular non-commercial partners (sweethearts)’. This finding is not found in the results section and appears for the first time in the discussion section. It is also not clear from the study methodology how did the authors arrive at this finding. 

RESPONSE: We thank the reviewer for raising these valid concerns. We have included the following statement to explain sweetheart in the context of this study in the Introduction section (Lines 87-90):

In the Cambodian context, “sweethearts” refer to romantic non-commercial sexual relationships. For FEWs, a sweetheart is typically a boyfriend or regular client in the form of indirect transactional sex through living support, dinner dates, gifts, or shopping trips. Such relationships usually involve a lack of condom use as displays of trust and intimacy.

We also revised the wording and added references to support the statement on Lines 323-324 in the Discussion.

In our recent study, unprotected sex was reported by FEWs to be a way to express trust and faithfulness to their regular non-commercial partners (sweethearts) [11, 19].

4. The authors mention that they conducted HIV testing for those who participated in the survey. However, it is not clear what happened with those who already knew their HIV status. Were they retested for this study? More information on this needs to reported in the methods and results section. The only clue regarding this comes from lines 301-303 in the discussion section, where the authors mention that most participants older than 30 years knew their HIV status. Information about ART status of those who knew their status will also be helpful. 

This national integrated biological and behavioral survey aimed to track the changes in the prevalence of HIV and risk behaviors among female entertainment workers, an HIV key population in Cambodia. Therefore, all participants received HIV testing, followed by the questionnaire interviews. In the questionnaire, we also asked participants about their HIV status. Those who already knew their HIV-positive status were further questioned about HIV treatment and care services they had received. We have revised the paragraph as follows (Lines 189-201):

All participants received HIV testing regardless of whether they already knew their HIV-status. A blood sample was obtained from each participant by a trained laboratory technician through finger-prick and tested for HIV antibodies using the HIV-1/2 Determine™ test, following the national protocol [27]. We confirmed the test results by the HIV 1/2 STAT-PAK™ test on-site. If a specimen was reactive by the HIV-1/2 Determine™ test but non-reactive by the HIV 1/2 STAT-PAK™ test, the participant was recommended to go for a confirmatory test at an ART clinic of their choice with support from a community outreach worker. HIV test result was provided to the participants verbally after the interview, together with post-test counseling. All participants attended a pre- and post-test counseling session provided by trained counselors from HIV confidential counseling and testing centers located in the study sites. Those tested positive for HIV were referred to an ART clinic of their choice by a local NGO working in the respective area for care and treatment services according to the national guidelines. All participants received a token of appreciation valued at approximately US$4 for their time and effort.

We have also added ART status of those who knew their status in the Results (Lines 233-234).

5. The authors mention about desirability bias in the limitations section. They also mention that steps were taken to mitigate desirability bias. However, what steps were taken has not been mentioned. The only mention about this comes from line 340, which says “sufficient measure”. It is important to understand this, since, from the manuscript text, it seems that the surveyors, or at least the organisations that they worked for were known to the participants and the participants received some kind of services from them. This can affect participants’ response and hence it is important to explicitly mention what measures were taken to avoid this bias. It will also be helpful to mention who the surveyors were and how were they trained to avoid these biases. It would also be pertinent to mention how were the survey procedures were monitored. 

RESPONSE: We thank the reviewer for raising this critical issue that may lead to potential bias in the results and conclusions. The participant recruitment was done with support from NGOs and outreach workers working with female entertainment workers in the respective study sites. However, the questionnaire interviews were conducted by independent teams who did not work directly with female entertainment workers in HIV intervention programs.

We have provided further information regarding the sensitization and preparation for the data collection in ‘Data collection training’ as follows (Lines 151-163):

In this national survey, we involved representatives of key stakeholders, including FEW communities, in every stage of the survey, from study design to questionnaire development and validation, data collection, and finding dissemination. Pre-data-collection workshops and training were conducted with data collection teams and representatives of HIV key stakeholders, NGOs working with FEWs, and FEW communities. Data collection teams were composed of team leaders from the Surveillance Unit of the National Center for HIV/AIDS, Dermatology and STD, staff members of the respective provincial health department, and representatives of NGOs, community outreach workers, and FEW communities. The training covered: (1) sampling methods, including mapping and eligibility criteria, (2) informed consent procedures, (3) participants’ privacy and confidentiality, (4) study protocol and interview techniques, (5) blood specimen collection, and (6) record-keeping and completion of the survey forms. We also pretested the questionnaire during the training.

6. The authors report a higher odds of HIV among those with STI. However, the model used does not contain condom use frequency. Could this result be due to the fact that those with STIs have infrequent condom use, which could also be the independent risk factor for HIV? 

RESPONSE: Thanks for the opportunity to clarify this confusing result. Given the exploratory nature of this study, the multiple regression model was built based on the results from bivariate analyses. As described in the ‘Statistical analyses,’ we included age, education level, marital status, types of entertainment establishments and other variables that achieved p<0.05 in the bivariate analyses in the model simultaneously. We then removed variables not statistically significant from the model using a backward stepwise selection method. Condom use frequency with non-commercial partners was significantly associated with HIV infection in bivariate comparison but did not retain its significant association with HIV infection after controlling for other covariates. Thus, it was removed from the model.

7. In sample size calculations, the authors seem to have calculated the overall sample size required for inclusion in the survey. However, in the limitations (line 342), the authors mention that they were unable to reach the sample size for freelance FEWs. Were different sample sizes for different subgroups calculated? 

RESPONSE: Thank you for the comment. We did not calculate different sample sizes for different sub-groups. In this study, we regarded freelance sex workers as FEWs according to the definition employed in the national HIV program. Hence, the sample size was calculated for the entire study population. Despite meeting the overall target sample size, we acknowledged that the sub-group of freelance sex workers might be underrepresented, despite striving to reach this population. They are the most difficult to reach due to their hidden nature and illegality of sex work in the country.

8. In the methods section (line 179), the authors mention that participants received a, ‘gift card not exceeding 4USD’. Did participants receive different amounts? If yes, what was the criteria for who received what amount? 

RESPONSE: We apologize for the unclear statement. We have amended it for clarification as follows (Lines 200-201): 

All participants received a token of appreciation valued at approximately US$4 for their time and effort.

9. In 348, the authors use the term “recent history of STI symptoms”. While the definition of ‘recent’ is not mentioned in the methods, from the results section we understand that it is three months. Please mention this definition, along with any other relevant operational definitions, in the methods section. Also, please rethink the use of the word ‘recent’ in the discussion. 

RESPONSE: We agree that the description of the variables was not very clear. We have included the operational definitions in ‘Variables and measurements’ as follows (Lines 181-188):

For HIV risks, we collected information regarding the participant’s HIV status and sexual behaviors with commercial partners, defined as having sexual intercourse in exchange for money or gifts, and non-commercial partners. These included the number of sexual partners (continuous), condom use frequency (always, frequently, sometimes, never), STI diagnosis, and experiences of STI symptoms (yes or no) in the past three months. We also collected information on illicit drug use (heroin, marijuana, amphetamine-type stimulants, or other types of drugs) and alcoholic drinks at work (yes or no) in the past three months. For participants who reported living with HIV, we collected self-reported HIV care and treatment information, including ART. 

Throughout the text, we have refrained from using the word ‘recent’ but stated the exact time period for the individual variable for clarity.

10. In the conclusion, the authors recommend differentiated preventive services for FEWs in older age group since they had a higher odds of living with HIV than the lower age groups. I would request the authors to re-think about this. While it is true that the results show that FEWs had a higher proportion of those living with HIV. However, we do not know from the results how long ago did they test positive for HIV. In light of this, it is necessary that preventive measures should not be focussed only on the older age groups. It is necessary to ensure that preventive and testing services should be available from even the youngest age groups, since in this way, they might be able to effect behavioural change early, which might help prevent new HIV infections. 

RESPONSE: Thank you for raising this critical comment. We agree that our results did not tell when the infections occurred as we measured HIV prevalence, not incidence. It is likely that the association between HIV infection and older age instead explains the accumulative exposure to HIV infection of FEWs in the older age group, or they may have lived with HIV for years. We have revised the conclusions as follows (Lines 362-372):

This study documents the prevalence and risk factors of HIV infection among FEWs in Cambodia. The results showed that the HIV infection risk was associated with older age, the nature of the entertainment venues where FEWs were working, and history of illicit drug use and STI symptoms in the past three months. The risk of HIV infection was inversely associated with years of formal education attained. These findings indicate that differentiated strategies for HIV prevention among FEWs should be geared towards the FEWs working as freelancers or in veiled entertainment venues, such as massage parlors, and freelance FEWs. However, prevention efforts for venue-based FEWs should be maintained. Innovative interventions such as online services and mobile health technologies using text and voice messaging may be more useful than the current reliance on physical outreach activities to effectively reach freelance and other high-risk FEWs for early HIV detection and linking them to treatment and care services.

11. The authors mention that they found lower proportion of FEWs testing HIV-positive at the establishments. They also mention that, “street based FEWs tend to be older than those working in brothels and entertainment avenues”. Are these two facts related? Could it be that the entertainment avenues prefer to employ those who are younger? Or is it that if someone tests HIV-positive, then they might not be allowed to work at these entertainment avenues? Please revisit the related conclusion based on the answers to these questions. 

RESPONSE: Thanks for raising these excellent points, which are all true. We have revised the conclusions to reflect these facts and their relationship in the conclusions (Lines 362-372):

This study documents the prevalence and risk factors of HIV infection among FEWs in Cambodia. The results showed that the HIV infection risk was associated with older age, the nature of the entertainment venues where FEWs were working, and history of illicit drug use and STI symptoms in the past three months. The risk of HIV infection was inversely associated with years of formal education attained. These findings indicate that differentiated strategies for HIV prevention among FEWs should be geared towards the FEWs working as freelancers or in veiled entertainment venues, such as massage parlors, and freelance FEWs. However, prevention efforts for venue-based FEWs should be maintained. Innovative interventions such as online services and mobile health technologies using text and voice messaging may be more useful than the current reliance on physical outreach activities to effectively reach freelance and other high-risk FEWs for early HIV detection and linking them to treatment and care services.

12. The authors mention in the strengths that they had involved all stakeholders. It is very important during this kind of work to involve all the stakeholders from the beginning and I would like to congratulate the authors on this. It would be nice to understand a little more on what was done. 

RESPONSE: We thank the reviewer for these encouraging comments. In this national survey, we involved representatives of key stakeholders, including FEW communities, in every stage of the survey, from study design to tool development and validation, data collection, and findings dissemination. We have included this information in Materials and Methods (Lines 151-163).

In this national survey, we involved representatives of key stakeholders, including FEW communities, in every stage of the survey, from study design to questionnaire development and validation, data collection, and finding dissemination. Pre-data-collection workshops and training were conducted with data collection teams and representatives of HIV key stakeholders, NGOs working with FEWs, and FEW communities. Data collection teams were composed of team leaders from the Surveillance Unit of the National Center for HIV/AIDS, Dermatology and STD, staff members of the respective provincial health department, and representatives of NGOs, community outreach workers, and FEW communities. The training covered: (1) sampling methods, including mapping and eligibility criteria, (2) informed consent procedures, (3) participants’ privacy and confidentiality, (4) study protocol and interview techniques, (5) blood specimen collection, and (6) record-keeping and completion of the survey forms. We also pretested the questionnaire during the training.

13. In line 91-93, the authors mention, ‘The identified risk factors of the HIV infection in the study included being freelance sex workers and younger age of first sex (≤15 years)’. However, this factor seems missing from this study. Was this variable collected and analysed? 

RESPONSE: In this study, being freelance sex workers was significantly associated with HIV infection in both bivariate and multiple regression analyses. We have presented and discussed this finding in the Results, Discussion, and Conclusions. However, the national survey did not collect data on the age of first sex as it was not a significant factor in previous national surveys. We will include the question in our upcoming surveys of HIV key populations.

Minor comments 

1. Line 63 – The authors mention about a triphasic response. While, the second and third phases are clearly mentioned, the first phase is not clearly demarcated. It would be easier to read if the first phase is also clearly labelled so. 

RESPONSE: We apologize for this oversight. We have added ‘first phase’ to the second sentence in paragraph 1 of the introduction (Line 63).

2. Line 113 – It would be good to also mention the total number of provinces for readers who are not aware about Cambodian setting. 

RESPONSE: We have added the total number of provinces in Cambodia as follows (Lines 116-118):

This National Integrated Biological and Behavioral Survey among FEWs (FEW-IBBS) was conducted in 2016 in Phnom Penh and 17 other provinces purposively selected out of the 25 provinces in Cambodia.

3. Line 203 – It might be pertinent to report on how many participants refused to participate. 

RESPONSE: We have added the following information at the beginning of the Results section (Lines 228-231):

Of 3353 FEWs invited, 148 (4.4%) refused to participate in the study, primarily due to their time constraints. Fifty-six participants (1.8%) with missing data of main variables or HIV testing results were further excluded from the analyses. In total, we included 3149 FEWs in the analyses. The mean age of the participants was 26.2 years (SD 5.7).

4. The tables as they stand are not very clear which makes them difficult to read. Please format them in a manner so that the subgroups and the totals are easily readable. Also, using row percent might make the tables easier to interpret, especially for tables 1 and 2. 

RESPONSE: We thank the reviewer for this comment. We understand that there are different ways to present the tables. However, we have received different opinions from reviewers and our research team members. Therefore, we would like to keep the table presentation and format in the way they are now. We believe these are also a common way employed in scientific papers. We hope the reviewer would agree with this.

---

## [Editor Report · Decision Letter 1]

9 Dec 2020

Risk factors of HIV infection among female entertainment workers in Cambodia: findings of a national survey

PONE-D-20-22691R1

Dear Dr. Yi,

We’re pleased to inform you that your manuscript has been judged scientifically suitable for publication and will be formally accepted for publication once it meets all outstanding technical requirements.

Kind regards,

Petros Isaakidis

Academic Editor

PLOS ONE
---

## [Editor Report · Acceptance letter]

11 Dec 2020

PONE-D-20-22691R1 

Risk factors of HIV infection among female entertainment workers in Cambodia: findings of a national survey 

Dear Dr. Yi:

I'm pleased to inform you that your manuscript has been deemed suitable for publication in PLOS ONE. Congratulations! Your manuscript is now with our production department. 

Kind regards, 

on behalf of

Dr. Petros Isaakidis 

Academic Editor

PLOS ONE